# Electrolytic Characteristics of Microhole Array Manufacturing Using Polyacrylamide Electrolyte in 304 Stainless Steel

**DOI:** 10.3390/mi14101808

**Published:** 2023-09-22

**Authors:** Junfeng He, Zan Wang, Wenjie Zhou, Yue Jian, Li Zhou

**Affiliations:** 1School of Mechatronic Engineering, Guangdong Polytechnic Normal University, Guangzhou 510665, China; hejunfeng@gpnu.edu.cn (J.H.);; 2Interdisciplinary Research Institute, Guangdong Polytechnic Normal University, Guangzhou 510450, China

**Keywords:** polyacrylamide electrolyte, non-Newtonian fluid, masked electrochemical machining, microhole

## Abstract

Because of the ease with which oxide films form on its surfaces, stainless steel has strong corrosion resistance and excellent processing performance. Electrochemical machining (ECM) is a flexible process that can create microstructures on stainless steel (SS304); however, with traditional masked ECM, the efficiency and accuracy of microstructure machining are low. Proposed here is the use of a non-Newtonian fluid [polyacrylamide (PAM)] as the electrolyte. To date, there have been few papers on the electrochemical dissolution behavior of stainless-steel micromachining with a non-Newtonian fluid as the electrolyte. The aims of the study reported here were to investigate the electrochemical properties of SS304 with PAM and PAM–NaOH as electrolytes, and to explain their electrochemical corrosion mechanisms. The effects of different electrolytes were compared, and the polarization curves of SS304 in PAM and PAM–NaOH electrolyte solutions with different components were analyzed and compared with that in NaNO_3_ electrolyte. Then, the effects of the main processing parameters (pulse voltage, frequency, and duty ratio) on the machining performance were investigated in detail. A microhole array was obtained with a good quality comprising an average diameter of 330.11 µm, an average depth of 16.13 µm, and a depth-to-diameter ratio of 0.048. Using PAM to process microstructures on stainless-steel surfaces was shown to be feasible, and experiments indicated that the mixed electrolyte (PAM–NaOH) had not only the physical characteristics of a non-Newtonian fluid but also the advantages of a traditional electrolyte to dissolve processing products, and it effectively improved the processing accuracy of masked ECM for SS304.

## 1. Introduction

Because of the ease with which oxide films form on its surfaces, 304 stainless steel (SS304) has strong corrosion resistance and excellent performance [1,2]. Electrochemical machining (ECM) is a manufacturing method in which a redox reaction is used to solvent and remove material from a workpiece to achieve machining. ECM is a good-quality and economic method, especially for microfabrication [3,4]. However, the characteristics of stainless steel make material removal difficult using traditional ECM methods.

Bian et al. [5] studied five different materials for the working cathode of ECM, using stray-current corrosion, taper, and material removal rate as the evaluation criteria for the drilling quality and efficiency of a thin-wall workpiece made of SS304. Luo et al. [6] reported the passivation and electrochemical behavior of 316 L stainless steel (SS316L) in chlorinated simulated concrete pore solutions at different pH values as evaluated by potentiodynamic measurements and electrochemical impedance spectroscopy. Qi et al. [7] investigated the influence of nitrogen alloying on the corrosion resistance of martensitic stainless steels with different nitrogen contents by immersion corrosion measurement and electrochemical measurement in a chloride medium. Freire et al. [8] studied the passivation and passivation breakdown of AISI 304 stainless steel in alkaline solutions with different pH values (from 13 to 9), simulating the interstitial concrete electrolyte; the results showed that pH plays an important role in the evolution of film resistance and charge transfer, and the effect is highly dependent upon the chloride content and immersion time.

In ECM, the electrolyte facilitates the anode dissolution and ion mass transfer of the workpiece; it not only facilitates electrolysis but also removes electrode products, greatly affecting the processing rate and accuracy [9,10,11]. Different electrolytes have different chemical components and machining capacities, possibly leading to differences in workpiece machining quality and efficiency. Man et al. [12] characterized and compared the structure and composition of primary and secondary passive films formed on AM355 martensitic stainless steel in NaOH. Patel et al. [13] presented a way to reduce the pulse energy per unit time by changing the waveform of the voltage pulse to reduce the overcut in NaNO_3_. Sometimes the workpiece is exposed to a low-current-density electric field (<5 A/cm^−2^), resulting in undesirable dissolution. Wang et al. [14] showed the electrochemical dissolution behavior of SS304 and Inconel 718 at low current density in NaNO_3_ electrolyte solution; the results indicated that the surface of SS304 is protected well by a compact oxide film that forms in NaNO_3_ solution at low current density. Wang et al. [15] also reported the electrochemical dissolution behavior of S-04 high-strength stainless steel in NaNO_3_ aqueous solution. Nie et al. [16] investigated the pitting and passivation of additively manufactured SS316L in 3.5 wt.% NaCl solution in comparison to its wrought counterpart; the results showed that additively manufactured SS316L had a fine subgrain structure with a diameter of ca. 5 μm, with the subgrain boundaries enriched by dislocations and molybdenum. Liu et al. [17] investigated the influence of passivation on the ECM performance, electrochemical behavior, and passive film properties of S136H steel in neutral solutions composed of either NaClO_3_ or NaNO_3_; the results showed that, compared with using the NaNO_3_ electrolyte, using the NaClO_3_ electrolyte offered better material removal rate and profile accuracy of microstructures on S136H steel.

Localization is a major factor in evaluating manufacturing accuracy in ECM, and a composite electrolyte solution helps to improve the processing localization of stainless steel [18,19,20]. Thanigaivelan et al. [21] compared the performances of acidified sodium nitrate and sodium nitrate in electrochemical micromachining, and the results showed that the machining rate and overcut improved significantly when acidified sodium nitrate was used as the electrolyte. Lo et al. [22] investigated the effect of H_2_SO_4_/HCl composition on selective dissolution, finding that it could occur in duplex stainless steels because of the difference in chemical composition between the two constituent phases. Kumaravel et al. [23] conducted machining of austenitic SS316L using plain aqueous NaOH and nitrogen-gas-assisted aqueous NaOH electrolytes to prevent surface cracks, heat-affected zones, and surface irregularities. Chen et al. [24] added a certain amount of disodium ethylene diamine tetra acetate (EDTANa_2_) to NaClO_3_ solution as the electrolyte, and they processed micropores on SS304 with a thickness of 400 μm; adding EDTANa_2_ reduced the deposition of electrolytic products on the tool electrode and improved the processing stability.

All of the aforementioned studies indicate that ECM with special electrolyte is a flexible process capable of creating microstructures in stainless steel. However, in traditional ECM, the machining efficiency and localization of microstructures are low; thus, improving the precision and stability of ECM is an important research project for stainless-steel machining. The dielectric characteristic of the electrolyte is a predominant parameter determining the performance of ECM, and proposed herein is the use of a non-Newtonian fluid [polyacrylamide (PAM)] as the electrolyte. To date, there have been few reported studies of the electrochemical dissolution behavior of stainless-steel micromachining with a non-Newtonian fluid as the electrolyte. As a polymer electrolyte solution, PAM has higher current density and viscosity than those of traditional electrolyte solutions; hence, substantially improved localization and precision are expected. To study the electrochemical properties of SS304 with PAM and explain its electrochemical corrosion mechanism, the effects of different electrolytes are compared, and the polarization curves of SS304 in PAM and PAM–NaOH electrolyte solutions with different components are analyzed and compared with those for NaNO_3_ electrolyte.

## 2. Materials and Methods

### 2.1. Experimental Device

The process of masked ECM (MECM) with PAM electrolyte is shown schematically in Figure 1, along with photographs of the system used in the experiments. A three-coordinate micro-displacement platform was used as the experimental device. A rotating disc electrode was selected as the tool electrode; made of brass, it had a diameter of 5 mm.

Species transport and current density are related mainly to the polar groups in the working fluid, and using a rotating disc electrode offers an increased steady-state diffusion mass transfer rate and a more uniform current density. Because of the rotation of the electrode, the air bubbles generated by processing are more likely to accumulate near the edge of the disc, thereby reducing their impact on the processing area. At the same time, the rotating disc electrode continuously updates the electrolyte in the processing gap, thereby keeping the conductivity in the gap within a certain range and, thus, reducing the instability of the relationship between current efficiency and current density.

During the machining, the rotating disc electrode was the cathode, and the workpiece was the anode. The workpiece was fixed onto an insulated plate, and the nozzle was fed in the *Z*-direction to contact the workpiece. The electrolyte jetted from the nozzle and reached the workpiece surface through a microhole (MH) array in a mask, whereupon an MH array could be prepared on the workpiece via electrochemical dissolution by controlling the movement of the workpiece at a certain speed. The charged electrolyte was sprayed continuously onto the workpiece area waiting to be processed through the disc electrode so that the workpiece material was etched selectively. The workpiece moved according to a planned path of the micro-displacement platform until all the MHs had been processed. The industrial computer used to control the experimental device system was developed by the Mitsubishi Corporation, and the electrolyte circulation system included a reservoir, a high-pressure pump, a filter, valves, a pressure meter, and several pipes.

An SOYI-20050DM pulsed power supply was used, and the fixture and electrolysis cell were made from acrylic. The surface morphology and dimensional accuracy were analyzed using a confocal laser scanning microscope (OLY4000; OLYMPUS, Tokyo, Japan), a hyperdepth 3D microscopy system (VHX-600E; KEYENCE, Shanghai, China), a Raman spectrometer (LabRAM HR Evolution; HORIBA JY, Palaiseau, France), a rotational rheometer (HAAKE Mars60; Thermo Fisher Scientific, Waltham, MA, USA), a Zennium electrochemical workstation (ZAHNER, Kronach, Germany), and a scanning electron microscope (S-3400N; Hitachi, Tokyo, Japan).

In the study reported herein, non-ionic PAM was chosen, and the experimental workpieces were SS304 sheets with a size of 50 mm × 50 mm × 1 mm; see Table 1 for the chemical composition of SS304. Photoresist masks were made on the workpiece surface before machining; the photoresist type was SU-8 with an adhesive thickness of 50 µm, and the template patterns were arrays of micropores.

### 2.2. Electrolyte

PAM is a water-soluble high-molecular-weight polymer formed by free-radical polymerization of monomer acrylamide. The chemical formula of PAM is (C_3_H_5_NO)*_n_*, and its density is 1.30 g/cm^3^ at 23 °C. PAM has a white powdery solid structure at room temperature which shown in Figure 2, and, according to its ionic properties, it can be divided into four types: nonionic, anionic, cationic, and zwitterionic.

PAM is readily soluble in water to form a non-Newtonian fluid that exhibits the deformability and softness of an elastic solid. Meanwhile, the high water content of PAM hydrogels results in them having liquid-like properties, including permeability to various chemical and biomolecules and transparency to optical and acoustic waves [25,26]; see Table 2 for the common physical properties of PAM.

PAM is insoluble in most organic solvents, and its aqueous solution does not degrade significantly until temperatures below 110 °C. At a suitable low concentration, PAM aqueous solution can be regarded as a network structure, and the mechanical entanglement of and hydrogen bonds between the chains form a network of nodes that can be used for aldehydes (such as formaldehyde) and high-valency metals (such as aluminum, chromium, zirconium, etc.), which are cross-linked by polynuclear hydroxyl bridge ions and degraded by either mechanical action or oxidation.

PAM aqueous solution is highly compatible as an electrolyte and is insensitive to strong acids and bases. The presence of double bonds and amide bonds in acrylamide means that hydrolysis, acylation, and enolization of amide groups occur during the polymerization process; chain transfer of chain radicals produces crosslinking, meaning that the PAM polymer structure contains branched chains and imines. This polymerization reaction conforms to the general rules of free-radical polymerization reactions, including free-radical chain initiation, growth, transfer, and termination. The polymerization reaction and chain initiation are described chemically as follows:(1)$$\begin{array}{llll} {\mathrm{CH}}_{2}=CHCONH_{2}\overset{Initiator}{\to }-CH_{2}- & CH-CH_{2}- & CH-\\ & | & | & \\ & CONH_{2} & CONH_{2} \end{array}$$
(2)$$\begin{array}{l} \begin{array}{lll} {\mathrm{CH}}_{2}=CHCONH_{2}+R-CH_{2}- & CH-CH_{2}- & CH-\\ & | & |\\ & CONH_{2} & CONH_{2} \end{array}\\ \begin{array}{lllll} \to R-CH_{2}- & CH-CH_{2}- & CH-CH_{2}- & CH-CH_{2}- & CH-\\ & | & | & | & |\\ & CONH_{2} & CONH_{2} & CONH_{2} & CONH_{2} \end{array}\\ \begin{array}{llll} & & & H\\ & & & |\\ \overset{+nCH_{2}=CHCONH_{2}}{\to }R-CH_{2}- & CH-[{CH}_{2}- & CH{]}_{n}-CH_{2}- & C\\ & | & | & |\\ & CONH_{2} & CONH_{2} & CONH_{2} \end{array} \end{array}$$

In the electrochemical reaction, the covalent bonds between molecules are broken, and an ion is formed when all electrons are captured by one of the atoms; if the split makes the shared electron pair belong to two atoms (or groups), then a free radical is formed.

PAM contains many polar groups (–CONH_2_) after hydrolysis and, thus, easily reacts chemically with other substances. In the PAM solution, the iron ions on the surface of the workpiece react to form Fe^2+^ and Fe^3+^, which then enter the electrolyte. At the same time, they react with the polar groups hydrolyzed by PAM to form a metal complex in a colloidal state, and the electrolysis rate increases with the concentration of polar groups:(3)$$\begin{array}{lll} {\mathrm{CH}}_{2}= & CH+2H_{2}O\overset{\mathrm{hydrolyzation}}{\to }-CH_{2}- & CH+NH_{4}^{+}+OH^{-}\\ & | & |\\ & CONH_{2} & COOH_{} \end{array}$$
(4)$$\begin{array}{lll} 2-[{\mathrm{CH}}_{2}- & CH{]}_{n}-+Fe\to (-[CH_{2}- & CH{]}_{n}-{)}_{2}Fe+H_{2}↑\\ & | & |\\ & COOH & COO \end{array}$$
(5)$$\begin{array}{lll} 4(-[CH_{2}- & CH{]}_{n}-{)}_{2}Fe+4-[CH_{2}- & CH{]}_{n}-+O_{2}={\left(-[CH_{2}-CH]-\right)}_{3}Fe+2H_{2}O\\ & | & |\\ & COO & COOH \end{array}$$

Fe^2+^ and Fe^3+^ generated in the anode reaction also react with the OH^−^ ions produced by hydrolysis to precipitate Fe(OH)_2_ and Fe(OH)_3_. In the electrochemical reaction, the PAM electrolyte also contributes to the coagulation, adsorption, and precipitation of Fe(OH)_2_ and Fe(OH)_3_ generated by the reaction. These steps speed up the elimination of electrolytic products and further improve the processing efficiency. This means that PAM can be used to process SS304 effectively by electrolytic machining. PAM is inexpensive and is a water purification material itself; hence, it is an environmentally friendly electrolyte.

### 2.3. Experimental Design

To explore the electrolytic properties of PAM, three experiments were conducted in this study. The first involved measuring polarization curves; the electrochemical corrosion polarization curves of SS304 in different concentrations of PAM, NaNO_3_, and PAM–NaOH electrolytes were measured using an electrochemical workstation (Zennium E). Their passivation and electrochemical corrosion abilities were compared, and then the electrolyte was optimized. The second experiment involved analyzing the corrosion morphology; the dissolution of SS304 was evaluated according to the corrosion morphology and energy spectrum at a representative potential selected from the polarization curve results, and the electrochemical corrosion mechanism of SS304 was elucidated. The third experiment involved a set of single-factor experiments involving machining an MH array on an SS304 sheet; the effects of pulse voltage, frequency, and duty ratio on the average depth, lateral erosion quantity, and depth-to-diameter ratio were evaluated in MECM, and then the experimental parameters were optimized to obtain an MH array with good comprehensive quality. The experimental parameters for the viscosity test are given in Table 3, The parameters for the polarization curve test are given in Table 4, and the parameters for the single-factor experiments are given in Table 5. Each group of single-factor experiments was performed three times, and the average of the experimental results was taken for further analysis.

The workpiece was inspected after machining, with 30 MHs selected randomly for each workpiece, and the MH array characterization data for a given set of experimental parameters was taken as the average of the data for these 30 MHs. A confocal laser scanning microscope (LEXT OLS4000; OLYMPUS) was used to obtain the three-dimensional morphologies and dimensions of the MHs, and a scanning electron microscope (S-3400N; Hitachi) was used to record their surface morphologies.

The MH structure is shown schematically in Figure 3. The MH diameter and depth are *D_t_* and h, respectively, the mask aperture is *D*_0_, and the characterization factors of the MH array are outlined below. The processing path for MH structure is shown in Figure 4.

Dimensional accuracy was evaluated from the average diameter (*D*) and average depth (*H*) of the MH array, i.e.,
(6)$$D=\frac{\sum_{i=1}^{n}D_{i}}{n},$$
(7)$$H=\frac{\sum_{i=1}^{n}h_{i}}{n},$$
where *D_i_* and $h_{i}$ are the diameter and depth, respectively, of the *i*-th MH, and n is the number of MHs detected. Shape accuracy is expressed mainly by the depth-to-diameter ratio (I) and the lateral erosion quantity (*D_v_*), which reflects the lateral corrosion degree of the MH array; the smaller the value of *D_v_*, the smaller the lateral erosion of the MH array, indicating excellent localization. In addition, in practice, an MH array with larger I is better. The expressions for *D_v_* and *I* are
(8)$$D_{v}=\sum_{i=1}^{n}\frac{D_{ti}-D_{0}}{2},$$
(9)$$I=\frac{H}{D}.$$

## 3. Results and Discussion

### 3.1. Viscosity of PAM Electrolyte

Non-Newtonian fluids are those that do not satisfy Newton’s law of viscosity, i.e., fluids that have a nonlinear relationship between shear stress and shear strain rate. As a typical non-Newtonian fluid, PAM also has unique physical properties. In MECM, the viscosity of PAM determines the distribution of the flow field and the air-bubble confinement ability. The viscosity of PAM electrolyte with different concentrations was measured using a rotational rheometer (HAAKE Mars60; Thermo Fisher Scientific).

As shown in Figure 5, the viscosity of the PAM electrolyte is much higher than that of traditional electrolyte, and, when the shear rate increases, the viscosity of the PAM electrolyte decreases, which is the phenomenon of shear thinning. The results also show that PAM has the physical characteristics of a non-Newtonian fluid. Because rotating the spindle during processing drives fluid movement between the tool electrode and the workpiece, the viscosity decreases locally, which also facilitates the discharge of electrolysis products.

### 3.2. Electrochemical Dissolution Characteristics

The electrochemical performance of SS304 in PAM electrolyte was studied using PAM (non-ionic), PAM–NaOH, and NaNO_3_ as electrolytes. The test involved a three-electrode system, with the reference electrode being a platinum sheet with the dimensions of 20 mm × 20 mm × 0.1 mm. The potential scanning range of the polarization curve was from −2 V to 4 V, the scanning speed was 50 mV/s, and the voltage amplitude was 10 mV.

As shown in Figure 6a, the polarization curve contained an obvious passivation zone, indicating that the time taken to break through the oxide layer on the workpiece was longer in the PAM solution, and the corrosion rate in this zone was relatively low. On the polarization curve for a molecular weight of 8 million, when the voltage exceeded 1 V, the current density increased rapidly and then flattened with a further increase in the potential; this is because overpassivation occurred at this time, the oxide film was broken down, and the anode material began to be removed. On the polarization curve for a molecular weight of 1800 million, the curve changed significantly when the voltage was close to 2 V, after which the current density decreased with increasing voltage.

The electrolysis rate increased with the concentration of polar groups (–CONH_2_), the formation of Fe^2+^ and Fe^3+^ increased with increasing current density, and the generated metal complexes also increased. When a passivation film forms on the surface, it hinders the electrolytic reaction. Experiments have shown that, compared to electrolytes without passivation, better dimensional accuracy and surface finish can be obtained with a passivation transition during electrolyte machining.

As shown in Figure 6b, the current density (*lgi*) value of the passivation interval with different concentrations of PAM electrolyte was smaller. The threshold passivation potential did not change with concentration. A 2–5 wt.% concentration range of PAM electrolyte (the interval for each group was 1%) was preferred for further optimization. Generally speaking, the *lgi* value in the passivation interval increased gradually in this electrolyte, and the polarization curve did not significantly change with increasing concentration; the minimum *lgi* value was 8.478 × 10^−5^ A/cm^2^ at 2 wt.% in the passivation interval. According to the principle of electrochemical corrosion mechanism and the processing efficiency, 2 wt.% PAM electrolyte was selected as the preferred electrolyte.

Figure 7 shows the corrosion morphologies of SS304 surface in PAM electrolyte at 2 wt.%. Generally, when the activation potential was applied, the workpiece could corrode. At 3.0 V, there was pitting on the surface of the SS304, and, as time passed, the pits gradually expanded and became contiguous, forming a larger corrosion area.

As shown in Figure 8, the Raman spectrum of PAM before and after processing showed two characteristic peaks. The peak at ca. 560 cm^−1^ represented the C–H, and the peak at ca. 1091 cm^−1^ represented the alcohol functional group (hydroxyl, –OH) in the solution. Because of the carbonyl C=O, the oxygen atom has stronger electronegativity and greater attraction to electrons. This shows that PAM can undergo enolization. Although PAM has many polar groups, as an electrolyte, it has obvious stability before and after being used in electrolytic processing.

Because of the high viscosity of PAM, the processing products cannot easily exit the processing area; thus, NaOH was added to dissolve them. Figure 9 shows the polarization curves for the electrochemical corrosion of SS304 in PAM, NaNO_3_, and PAM–NaOH as different electrolytes, where PAM–NaOH had a higher self-corrosion potential and activation potential than NaNO_3_. Figure 9 shows that, with the traditional electrolyte (NaNO_3_), there was no obvious passivation area, and the overpassivation potential was relatively large. When using PAM, the polarization curve showed a significant passivation threshold; hence, PAM is a passivating electrolyte. The self-corrosion potential of the PAM–NaOH electrolyte was maintained at −0.75 V, and the passivation area was slightly longer than that with PAM. The potential of threshold passivation was maintained at ca. 1.0 V, and, in the superpassivation stage, *lgi* increased rapidly with increasing potential.

Electrolyte concentration is a key factor influencing ECM. Accordingly, 1 wt.% PAM and 1 wt.% NaOH were mixed into a 1:1 PAM–NaOH solution, and a total of seven mixed electrolytes with different ratios were produced using the same method. Figure 10a shows that, with an increasing mixing ratio of PAM–NaOH, *D_v_*, showed a fluctuating upward trend. This may be because, after NaOH was added to the electrolyte, the number of OH^−^ ions in the electrolyte increased; OH^−^ could dissolve the machining products, making it easier for the electrolyte to react with SS304. The amide group of PAM became more and more active, the ability to take away metal ions was enhanced, and the increase in current density increased the material removal rate. As Figure 10b shows, *I* also increased gradually; however, considering the viscosity characteristics of PAM, too much NaOH is not conducive for renewing PAM in the processing area. Therefore, 1:2 was selected as the mixed electrolyte concentration for subsequent research.

### 3.3. Single-Factor Experiments

#### 3.3.1. Effects of Pulse Voltage

The pulse voltage is an important parameter in MECM; to ensure that the required current density is achieved, the magnitude of the voltage has an important influence. According to Faraday’s law, the dissolution rate is determined by the current density.

Figure 11a,b show that, as the voltage increased, *D_v_* and *H* increased with all three electrolytes, but *D_v_* of the pores processed with PAM and PAM–NaOH was smaller than that with NaNO_3_. This may be because, as the machining voltage increased, so did the current density in the machining area and the material removal ability, thereby increasin *D_v_* and *H* with all three electrolytes. A higher machining voltage led to a larger value of *D_v_*, a greater corresponding machining error, and worse machining accuracy. The precision of the MHs processed with PAM and PAM–NaOH electrolytes was improved, and the *D_v_* values with PAM–NaOH were the smallest. When the voltage was increased from 5 V to 35 V, *D_v_* increased from 10.35 μm to 23.59 μm, and *H* increased from 18.99 μm to 37.03 μm. Because the added NaOH could chemically react with the electrolytic products to dissolve them and avoid the concentration polarization phenomenon caused by product accumulation, the *D_v_* values of the MHs processed with PAM–NaOH were smaller than those with PAM. Figure 11c,d show the appearance of small holes processed under the action of 5 V and 35 V. The MH appearance under 35 V was poor, which may have been because of the violent reaction of the anode caused by the excessive voltage; products and air bubbles were difficult to remove in a timely manner, resulting in decreased accuracy. Figure 11e shows that *I* increased with increasing voltage and had a better processing effect with PAM–NaOH than with the other two electrolytes.

#### 3.3.2. Effects of Frequency

The frequency is another important parameter in MECM. As shown in Figure 12a,b, under the action of the three electrolytes, *D_v_* and *H* had the same change trends with increasing frequency, and the processing effect in PAM–NaOH was the best. As the frequency increased, *D_v_* decreased and the processing accuracy improved, but *H* also decreased. In the case of constant pulse width, increasing the frequency reduced the effective processing time in one cycle, and more processing products were removed. At the same time, adding NaOH dissolved the processing products, reduced their accumulation, and facilitated their discharge, thereby lessening the generation of air bubbles in the processing area. Rotating the tool constantly updated and replenished the electrolyte, which also improved the processing accuracy and decreased *D_v_*. When the frequency increased, the effective processing time in one cycle decreased, resulting in a reduced material removal rate and a decrease in *H*. Figure 12c,d show the surface quality of MHs under the action of 0.5 kHz and 3.5 kHz; as can be seen, the appearance of the MH under 3.5 kHz was better. Figure 12e shows that *I* decreased with increasing frequency and was the smallest in PAM–NaOH, as a function of the same reasons as above.

#### 3.3.3. Effects of Duty Ratio

Figure 13 shows the effects of the duty ratio. As shown in Figure 13a,b, both *D_v_* and *H* increased with increasing duty ratio, although *H* increased more slowly than *D_v_*; when the duty ratio exceeded 30%, the change tended to be flat. The main reason for the increases is that a larger duty ratio led to a longer effective processing time in one cycle, thus increasing the material removal rate, along with *D_v_* and *H*. As shown in Figure 13c,d, because the effective processing time was short when the duty ratio was small, there was sufficient time to discharge the products and renew the electrolyte; therefore, the processing quality was relatively high. However, when the duty ratio was increased, the reaction was stronger and the processing quality began to decline. Figure 13e shows that, with increasing duty ratio, *I* increased with all three electrolytes but was always lower with PAM–NaOH, which may have been because when the effective processing time was too short, the active ingredients in the PAM–NaOH electrolyte had no time to react with SS304. A small duty ratio reduced the machining efficiency of the workpiece.

Through the previous single-factor experimental analysis, the optimized experimental parameters were set as follows: the mixed electrolyte PAM–NaOH (1:2) was selected for processing, the machining gap was 1 mm, the aperture of the mask was 300 μm, the feed speed was 50 μm/s, the spindle speed was 500 rpm, the pulse voltage was 10 V, the frequency was 2.0 kHz, and the duty ratio was 30%. The representative MH morphology and contour processed with this group of experimental parameters are shown in Figure 14. The morphology of an MH array produced using these parameters was good.

## 4. Conclusions

In this study, PAM as a non-Newtonian fluid electrolyte was used for MECM of SS304 to study the electrochemical properties of PAM with SS304, and to explain the electrochemical corrosion mechanism of SS304. MECM was then used to explore the processing trends of an array of MHs on SS304, and the conclusions are summarized below.

By measuring and analyzing polarization curves and Raman spectra, the electrochemical characteristics of different electrolytes were compared. The polarization curves of SS304 in PAM and PAM–NaOH electrolyte solutions with different components were analyzed and compared with that in NaNO_3_ electrolyte, the electrochemical corrosion mechanisms of SS304 by PAM and PAM–NaOH electrolyte were elucidated, and the electrochemical reactions were summarized. Using PAM to process microstructures on stainless-steel surfaces was shown to be feasible.

Experiments indicated that, as the voltage and duty ratio increased, *D_v_* and *H* increased with all three electrolytes, but PAM–NaOH had the best processing effect. As the frequency increased, *D_v_* decreased and the processing accuracy improved, but *H* also decreased. Under the action of the three electrolytes, *D_v_* and *H* had the same change trends, but the processing effect in PAM–NaOH was again the best. This may have been because, after NaOH was added to the electrolyte, the number of OH^−^ ions in the electrolyte increased; OH^−^ could dissolve the machining products, making it easier for the electrolyte to react with SS304. Because of its non-Newtonian nature, the proposed electrolyte also offered better machining accuracy; the mixed electrolyte (PAM–NaOH) had not only the physical characteristics of a non-Newtonian fluid but also the advantages of a traditional electrolyte to dissolve processing products, and it effectively improved the processing accuracy of SS304 with MECM.

After optimizing the parameters, an MH array was obtained with a good quality comprising an average diameter of 330.11 µm, an average depth of 16.13 µm, and a depth-to-diameter ratio of 0.048.

## Figures and Tables

**Figure 1 micromachines-14-01808-f001:**
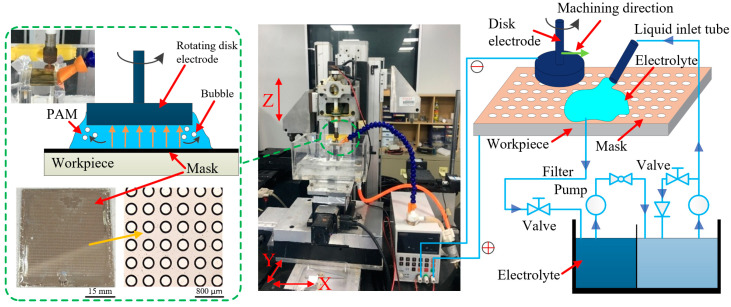
Photographs and schematics of masked electrochemical machining (MECM) with polyacrylamide (PAM) electrolyte.

**Figure 2 micromachines-14-01808-f002:**
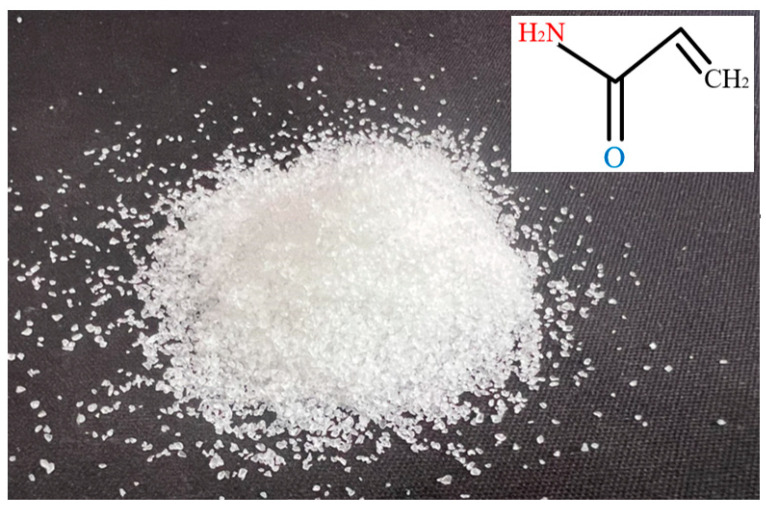
PAM and its molecular formula.

**Figure 3 micromachines-14-01808-f003:**
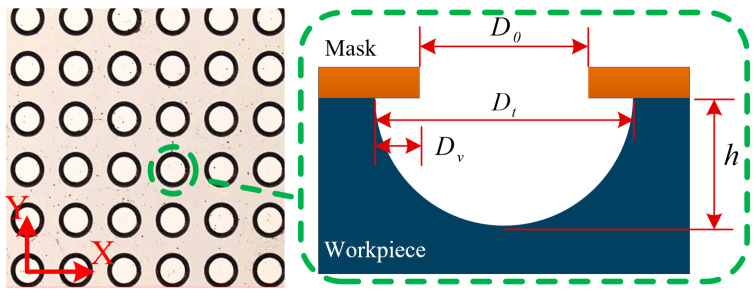
Schematic of microhole (MH) structure.

**Figure 4 micromachines-14-01808-f004:**
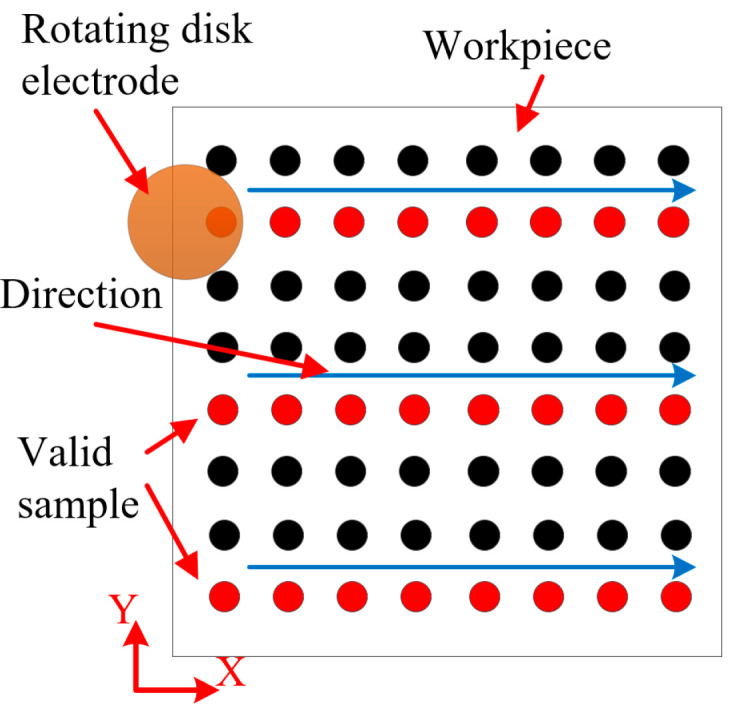
Processing path for MH structure.

**Figure 5 micromachines-14-01808-f005:**
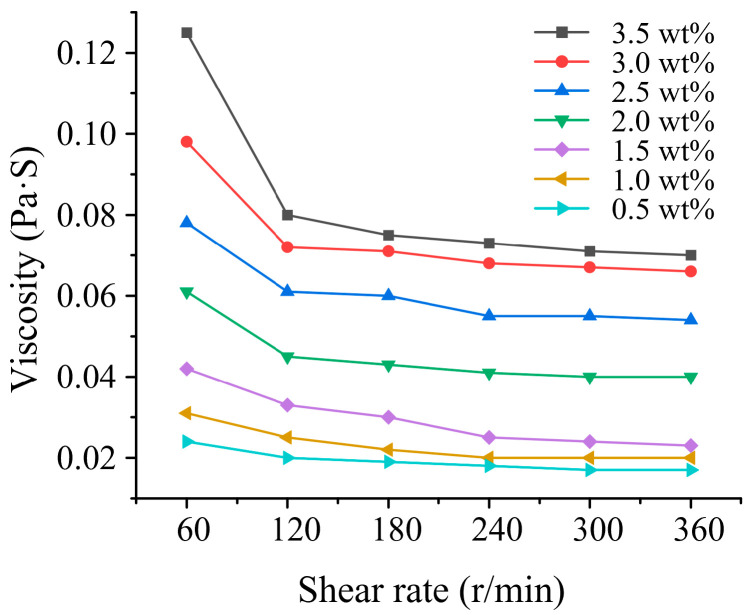
Viscosity of PAM electrolyte.

**Figure 6 micromachines-14-01808-f006:**
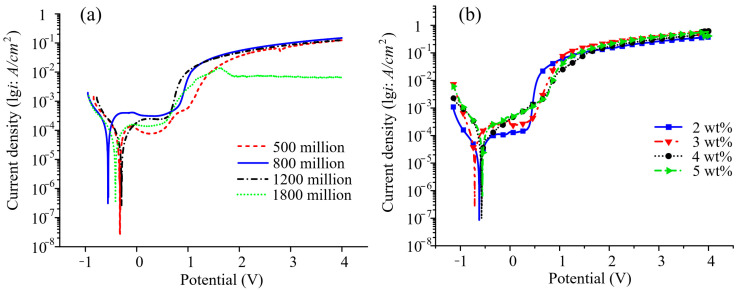
Electrochemical dissolution polarization curves for SS304 in PAM electrolyte: (**a**) different molecular weight (1 wt.%); (**b**) different concentration.

**Figure 7 micromachines-14-01808-f007:**
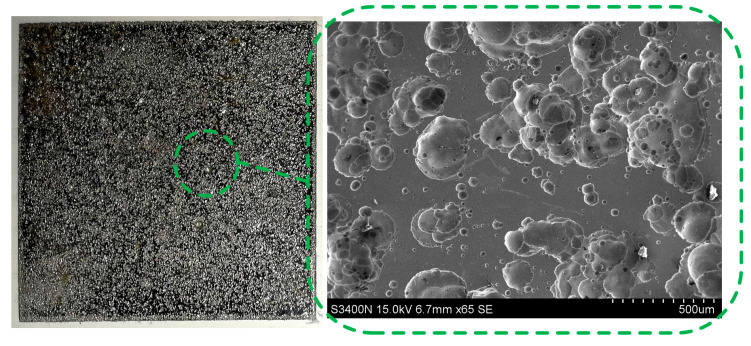
Surface corrosion morphologies of SS304 in PAM electrolyte (3 V).

**Figure 8 micromachines-14-01808-f008:**
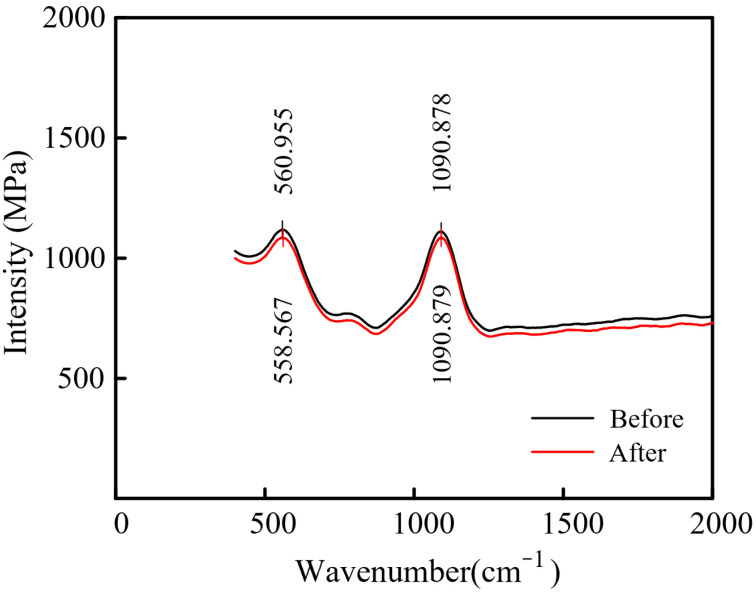
Raman spectrum of PAM before (black) and after (red) processing.

**Figure 9 micromachines-14-01808-f009:**
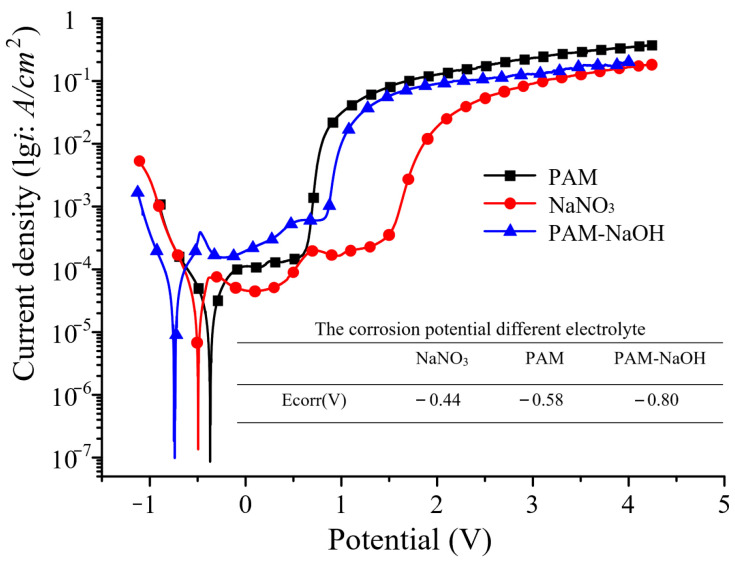
Electrochemical dissolution polarization curves in different electrolytes.

**Figure 10 micromachines-14-01808-f010:**
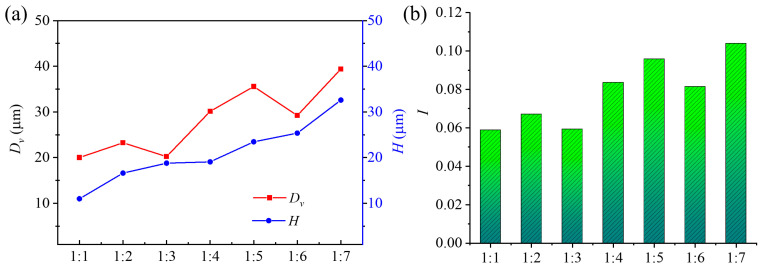
Effect of different proportions of PAM–NaOH solution on MHs (Other experimental parameters: pulse voltage 10 V, frequency 1.0 kHz, duty ratio 30%, free speed 500 rpm). (**a**) *D_v_*; *H*; (**b**) *I*.

**Figure 11 micromachines-14-01808-f011:**
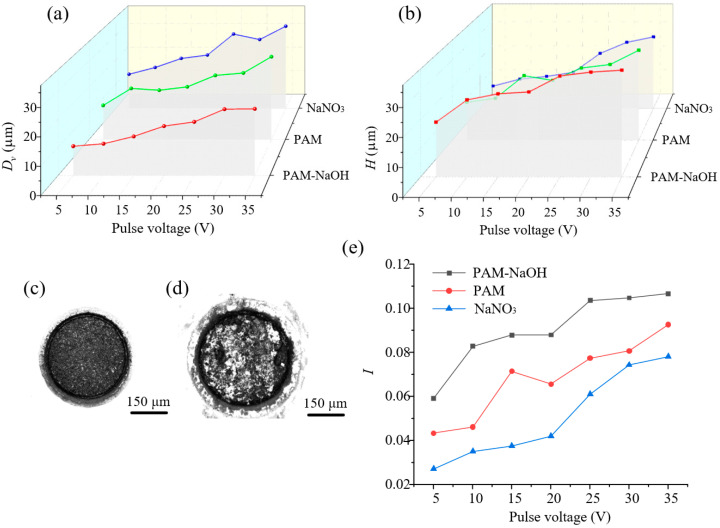
Results of MH machining with different values of pulse voltage (frequency = 1.0 kHz, duty ratio = 30%): (**a**) *D_v_*; (**b**) *H*; (**c**) appearance of MH under 5 V; (**d**) appearance of MH under 35 V; (**e**) *I*.

**Figure 12 micromachines-14-01808-f012:**
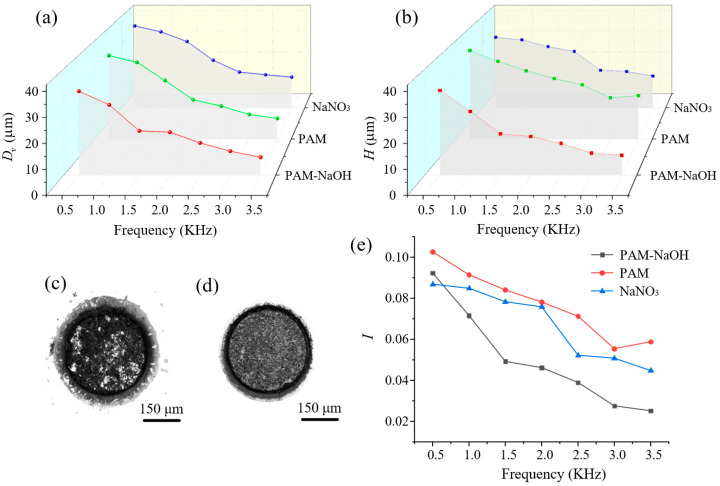
Results of MH machining with different value of frequency (voltage = 10 V, duty ratio = 30%): (**a**) *D_v_*; (**b**) *H*; (**c**) appearance of MH under 0.5 kHz; (**d**) appearance of MH under 3.5 kHz; (**e**) *I*.

**Figure 13 micromachines-14-01808-f013:**
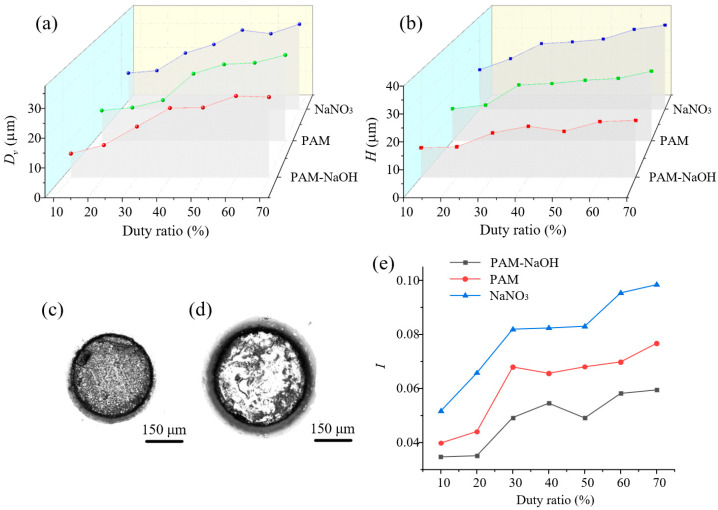
Results of MH machining with different values of duty ratio (voltage = 10 V, frequency = 2.0 kHz): (**a**) *D_v_*; (**b**) *H*; (**c**) appearance of MH under 10%; (**d**) appearance of MH under 70%; (**e**) *I*.

**Figure 14 micromachines-14-01808-f014:**
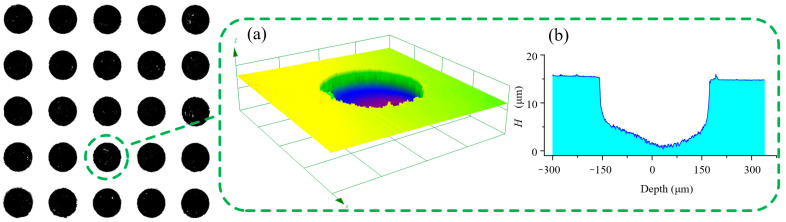
MH array processed using optimal parameters (an average diameter of 330.11 µm, an average depth of 16.13 µm, and a depth-to-diameter ratio of 0.048): (**a**) Morphology of MH; (**b**) Cross section of MH.

**Table 1 micromachines-14-01808-t001:** Chemical composition of SS304.

	C	Mn	P	S	Si	Cr	Ni	N
SS304 (wt.%)	0.08	2	0.035	0.015	0.75	18–20	8–10.5	0.1

**Table 2 micromachines-14-01808-t002:** Physical property of PAM at different concentrations.

Parameter	pH Value	Surface Tension [mN/m]	Viscosity [Pa∙s]	Conductivity [mS/cm]
1%	7.3	45.93	0.021	6.68
2%	7.1	32.3	0.035	9.54
3%	6.9	31.05	0.58	12.84

**Table 3 micromachines-14-01808-t003:** Experimental parameters for viscosity test.

Parameter	Value(s)
PAM [wt.%]	0.5, 1.0, 1.5, 2.0, 2.5, 3.0, 3.5
Shear rate [rpm]	60, 120, 180, 240, 300, 360
Time [s]	30
Pressure [kPa]	101.325
Temperature [°C]	25

**Table 4 micromachines-14-01808-t004:** Experimental parameters for polarization curve test.

Parameter	Value(s)
PAM [wt.%]	2, 3, 4, 5
NaNO_3_ [wt.%]	2
PAM:NaOH [wt.%]	1:1
Measuring potential [V]	−2 to 4
Scan rate [mV/s]	1
Temperature [°C]	25

**Table 5 micromachines-14-01808-t005:** Single-factor experimental parameters.

Parameter	Value(s)
Machining gap [mm]	1
Mask aperture [μm]	300
Concentration [wt.%]	2
Feed speed [μm/s]	50
Spindle speed [rpm]	500
Pulse voltage [V]	5, 10, 15, 20, 25, 30, 35
Duty ratio [%]	10, 20, 30, 40, 50, 60, 70
Frequency [kHz]	0.5, 1.0, 1.5, 2.0, 2.5, 3.0, 3.5

## Data Availability

Not applicable.

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
