# Peer review of "Electrolytic Characteristics of Microhole Array Manufacturing Using Polyacrylamide Electrolyte in 304 Stainless Steel"

_micromachines, 2023, doi:10.3390/mi14101808_

Round 1
Reviewer 1 Report
Title: Electrolytic Characteristics of Microhole Array Manufacturing Using Polyacrylamide Electrolyte in 304 Stainless Steel
Auth: Junfeng He, Zan Wang, Wenjie Zhou, Yue Jian and Li Zhou
Jour: micromachines​
Stainless steel exhibits robust resistance to corrosion along with exceptional processing capabilities. Electrochemical machining (ECM) offers a versatile method for creating microstructures on stainless steel (SS304). This submitted manuscript proposed a novel approach employing a non-Newtonian fluid, polyacrylamide (PAM), as an electrolyte, which aims to analyze SS304's electrochemical properties with PAM and PAM–NaOH electrolytes. Through this, high-quality microhole arrays with PAM were achieved, which enhances masked ECM precision on SS304. The viability of utilizing PAM for processing microstructures on stainless steel surfaces is effectively demonstrated through experimentation. As a result, the proposed method significantly enhances the precision of masked ECM when applied to SS304. On the whole, this manuscript is well completed, and the experimental workload is relatively large.The results which have great application value, can play a role in the fabrication of stainless steel microstructures. A few comments and questions are provided to the authors for consideration.
(I suggest consideration for publication after addressing the following minor comments:)
1. In line 14, there is a “]” lacking in “non-Newtonian fluid[polyacrylamide (PAM) as the electrolyte.” 2. In Table 1 on line 150, the chemical composition of SS304 is missing units. Whether the missing unit is a percent of mass fraction (wt%)? 3. In line 302, “the lgi value in the passivation interval increases gradually”. In Fig. 6(b), within the passivation interval, the data do not fully satisfy the rule that the higher the concentration, the greater the lgi value. For example, at the potential voltage of 1 V, lgi of 4 wt% is smaller than lgi of 3wt%. The statement in the text may not be rigorous. 4. In lines 302-303, “the minimum lgi value is 8.478×10−5A/cm2 at 2 wt%”. From Fig. 6, it seems that the minimum value is less than 8.478×10−5 A/cm2, whether there was a clerical error in the order of magnitude? 5. Figure 14 shows the morphology of the MH array processed using optimal experimental parameters. I wonder how many total samples were manufactured? Did the authors test all the microholes to verify the stability and accuracy of the fabrication method?Author Response
Thanks for your professional comments, and they are very helpful for improving our manuscript. According to your comments, we have revised our manuscript carefully, and revisions are listed as follows.

Reviewer 2 Report
Herein, the authors investigated the electrochemical properties of SS304 with PAM and PAM-NaOH as electrolytes and explain their electrochemical corrosion mechanisms. Then, the effects of the main processing parameters on the machining performance are investigated. Furthermore, this original work is well written and organized, but there are still some problems that should be revised. Minor revisions are needed here before it can be accepted for publication in Micromachines. The revised manuscript should be resubmitted after consideration of the following points:
1) In Figure 1, the scale bar should be added.
2) What are the processing efficiency of the proposed methods (PAM and PAM-NaOH as electrolytes)?
3) The SEM images of the fabricated microhole array must be added.
4) There are many excellent works maybe useful for the authors. Some important references in this field should be cited. Please refer “doi: 10.1002/adfm.202302311”, and “doi: j.surfcoat. 2023.129897”, and “doi: 10.1002/smll.202301745”.
Author Response
Thanks for your professional comments, and they are very helpful for improving our manuscript. According to your comments, we have revised our manuscript carefully, and revisions are listed as follows. Please see the attachment.

Reviewer 3 Report
The aims of the study reported here were to investigate the electrochemical properties of SS304 with PAM and PAM–NaOH as electrolytes and explain their electrochemical corrosion mechanisms. What is new in relation to current research is that the authors examined the effects of different electrolytes are compared, and the polarization curves of SS304 in PAM and PAM–NaOH electrolyte solutions with different components are analyzed and compared with those for NaNO3 electrolyte.
The structure of the article is correct. It contains an introduction, an analysis of the state of knowledge, a description of experiments, and an analysis of research results, which could be supplemented with a comparison of the results with other authors (if there are such results). Please check the correctness of formula (1-5) elements before publication.
Detailed comments below:
1. What is the applicability of proposed methods (PAM and PAM-NaOH as electrolytes)?
2. Figure 1 is missing scale
3. How were the variability ranges of the input parameters determined in tables 3-5?
4. In Figures 3 and 4, please add coordinate systems.
5. Figure 14 should be enlarged.
Minor editing of English language required
Author Response
Thanks for your professional comments, and they are very helpful for improving our manuscript. According to your comments, we have revised our manuscript carefully, and revisions are listed as followsm. Please see the attachment.
